# The Effects of a Partially Hydrolyzed Formula with Low Lactose and Probiotics on Mild Gastrointestinal Disorders of Infants: A Single-Armed Clinical Trial

**DOI:** 10.3390/nu13103371

**Published:** 2021-09-25

**Authors:** Yongying Huang, Yubo Zhou, Hongtian Li, Yipu Chen, Yingchao Mu, Anan Yuan, Yantao Yang, Jianmeng Liu

**Affiliations:** 1The Institute of Reproductive and Child Health/Ministry of Health Key Laboratory of Reproductive Health, Department of Epidemiology and Biostatistics, School of Public Health, Peking University Health Science Center, Beijing 100191, China; hyongying@126.com (Y.H.); zhouyubo@bjmu.edu.cn (Y.Z.); liht@bjmu.edu.cn (H.L.); 2Nestlé Product Technology Center-Nutrition, 1800 Vevey, Switzerland; Yipu.Chen@nestle.com; 3Huantai Maternal and Child Health Care Hospital, Zibo 256400, China; 13792199881@163.com; 4Nestlé (China) Ltd., Beijing 100102, China; Ann.Yuan@cn.nestle.com (A.Y.); Yantao.Yang@CN.nestle.com (Y.Y.); 5Center for Intelligent Public Health, Institute for Artificial Intelligence, Peking University, Beijing 100191, China

**Keywords:** partially hydrolyzed formula, low lactose, probiotics, mild gastrointestinal disorders, infant gastrointestinal symptom questionnaire, trials

## Abstract

Partially hydrolyzed formula (pHF) containing low lactose and probiotics may benefit the gastrointestinal health of infants. We aimed to assess the effects of pHF on mild gastrointestinal disorders (MGDs) of infants. In this single-armed trial, 80 full-term infants with MGDs were enrolled and fed a pHF for 14 consecutive days. The primary outcome resulted from the scores of gastrointestinal symptoms reported by parents using a validated Infant Gastrointestinal Symptom Questionnaire (IGSQ) at Day 0 (baseline), Day 7, and Day 14. The total IGSQ scores ranged from 13 to 65. Higher scores indicated worse gastrointestinal symptoms. The IGSQ scores (mean ± SD) decreased from Day 0 (36.0 ± 5.7) to Day 7 (28.7 ± 7.4) and Day 14 (26.5 ± 8.1 (*p* < 0.001), with corresponding digestive distress prevalence (IGSQ score > 30) decreasing from 87.5% to 35.0% and 28.8% (*p* < 0.001). In the first three days, vomiting and flatulence scores decreased at Day 1 versus Day 0, and the crying score decreased at Day 2, but no significant changes were observed for fussy and stool characteristics. All growth parameters increased and no parents reported adverse events. In conclusion, feeding with a pHF containing low lactose and probiotics may comfort infants with MGDs, and the comforting effect likely manifests early in the first three days of the feeding interventions. Trial registration: ClinicalTrials.gov NCT04112056

## 1. Introduction

Mild gastrointestinal disorders (MGDs) for infants, such as colic, regurgitation, diarrhea, and constipation, are not clinically diagnosed diseases, but occur commonly [1], discomforting infants, distressing families, increasing utilization in healthcare [1,2], or even affecting children’s long-term health [3]. Previous studies showed 27.8% of infants aged under 4 months old were affected by MGDs in Spain [1] and 42.6% for those aged under 6 months old in China [4]. The underlying mechanism is not well known [5]. Given that relevant symptoms do not suffice for medical treatments, MGDs are of much concern, particularly by parents, and, if not relieved timely, the concern is often further aggravated.

Without medically indicated treatments, dietary modifications appear appropriate to resolve these mild gastrointestinal symptoms [6]. Breast milk is likely an optimal option. The World Health Organization (WHO) has recommended to exclusively breastfeed newborns for more than 6 months [7], yet the breastfeeding rate is as low as 41% worldwide [8]. The rate of 29% in China is much lower, similar to many developed countries [9]. Consequently, cow’s milk-based formula is substituted for feeding newborns [10,11]. Newborns’ digestive systems are immature and vulnerable to ingredients contained in formula, such as proteins and lactose. Protein allergies and lactose intolerance frequently occur in formula-fed infants, likely leading to malabsorption-related MGDs. To feed vulnerable infants well, various formulas have been developed, such as partially hydrolyzed protein formula (pHF), low-lactose formula, or probiotics-containing formula [12,13,14]. These formulas were previously shown effective in relieving gastrointestinal symptoms in infants who were prematurely delivered or whose parents were allergic to milk protein [14,15,16]. pHF with low lactose was shown to be effective in relieving fussiness [17]. However, few studies have examined the effects of pHF with low lactose plus probiotics on MGDs.

We conducted a one-armed trial, where a pHF containing low lactose and probiotics was administered to full-term infants with MGDs for 14 consecutive days, to assess the before and after health effects on MGDs. Our hypothesis was that the pHF relieves mild gastrointestinal symptoms often occurring in formula-fed infants. 

## 2. Materials and Methods

### 2.1. Study Design and Subjects

This study was conducted in Huantai County, Shandong Province of China. The study was originally designed as a one-armed, before and after trial without randomized controls, due to ethical considerations that should not restrict participating infants to switch formula. Eligible infants were enrolled from October 2019 through March 2020 and individually allocated to be fed daily with the pHF containing low lactose and probiotics for 14 consecutive days. To be eligible, infants (1) were fed by formula exclusively or partially, and if fed partially, the formula accounted for over half of the daily diet; (2) were singletons born at ≥37 gestational weeks; (3) were from 7 to 180 days old; (4) were participating with parental consent; and (5) were generally healthy, but were reported by parents to be afflicted with MGDs. MGDs were defined as any one of the following symptoms: crying for three consecutive days and not easily comforted; moderate or severe flatulence, exhaust, or burp for three consecutive days; or difficult defecation, hard stools, or mushy stools that lasted for the past week. Difficult defecation was defined as defecating 3 times or less per week, pain with defecation, or large stools touchable on the abdomen. To be excluded, infants (1) were fed with the formula under study or new complementary foods in the past week, (2) took any medical drugs or any kind of probiotics, (3) were allergic to milk protein according to parents’ reports, or (4) had gastrointestinal syndromes that conformed to the Rome IV criteria for functional gastrointestinal disorders [5].

The study was registered at ClinicalTrials.gov (Identifier: NCT04112056), conducted in accordance with the Declaration of Helsinki, and approved by the Peking University institutional review board (IRB00001052-190103). Informed consents were signed by parents or legal guardians of infants before enrollment.

### 2.2. Study Formula

The study formula, compliant with the Good Manufacturing Practices (ISO 22000) and the Directive 2006/141/EC, was commercially manufactured. The study formula, similar to standard formula, was isocaloric (280 kcal/100 mL), containing protein, carbohydrate, fat, vitamins, minerals, and long-chain polyunsaturated fatty acids, but unlike standard formula, containing partially hydrolyzed proteins, low lactose of <3.1 g per 100 mL, and Bifidobacterium Bb12 of >106 CFU/g. The detailed compositions are shown in Appendix A. 

### 2.3. Outcome Measurements

The primary outcome resulted from the scores of gastrointestinal symptoms that were measured at Day 0 (baseline), Day 7 (±1), and Day 14 (±1) of feeding intervention by using a validated Infant Gastrointestinal Symptom Questionnaire (IGSQ) [18,19]. The IGSQ, containing 13 questions about infants’ gastrointestinal symptoms during the past week, was completed by parents under the guidance of trained health providers. The score for each of the 13 questions ranged from 1 to 5, leading to a total score of from 13 to 65. The 13 questions reflected 5 domains of gastrointestinal symptoms, including stooling (2 questions, scores: 2–10), vomiting (4 questions, scores: 4–20), flatulence (3 questions, scores: 3–15), crying (2 questions, scores: 2–10), and fussiness (2 questions, scores: 2–10). Higher scores indicated worse gastrointestinal symptoms. Digestive distress was defined as a total IGSQ score of >30 [18]. 

To examine whether the effects of feeding intervention manifested earlier, a simplified IGSQ containing 4 questions, i.e., times of vomiting, times of flatulence, duration of crying, times of fussiness in the past 24 h, was adopted to collect detailed information in the 3 days of feeding intervention. The score for each of the 4 questions also ranged from 1 to 5. Information on stool and sleeping in the first three respective days was also collected. Information on stool included frequency, pain, and consistency (watery, runny, mushy, firm, or hard); information on sleeping included sleeping hours and times of wake-ups at night. In addition, growth parameters for infants who wore indoor light clothing without shoes at Day 0 and Day 14 were measured by trained health providers, including length, weight, and head circumference. Body mass index (BMI, kg/m^2^) was calculated as weight in kilograms divided by squared height in meters. An age- and sex-specific Z-score of each parameter was calculated according to the WHO child growth standards [20]. At Day 14, parents were interviewed to answer the question, “are you satisfied with the feeding intervention”. For the question, there were three choices: so-so, satisfied, and un-satisfied. Throughout the study, adverse events or parents’ concerns were recorded and checked by health professionals. The flow chart is shown in Figure 1.

### 2.4. Quality Control

The operational manual was developed and all health providers involved in the study were trained by Peking University staff. A pilot was conducted to perfect field procedures and to familiarize the involved health providers with the procedures. Pilot data were excluded in the final analysis. During the study, health providers and parents were in close contact using a specially developed mobile application. Parents or guardians visited the designated hospitals at Day 0, Day 7, and Day 14 to complete, with health providers’ guidance, the IGSQ that assessed primary outcomes. For the visits at Day 7 and Day 14, parents were asked to show a picture of formula cans to help estimate the amount of formula consumed and to answer questions related to formula consumption and breastfeeding over the past 24 h. For the visits at Day 0 and Day 14, infants’ growth parameters were measured twice by the same health providers using the same equipment. The average values were used in the analysis. All completed questionnaires or measurements were checked online through the aforementioned mobile application. Through the online application, the relevant databases were built, and the progress of the study monitored in real time by Peking University staff, which enabled the fields to be timely informed, if procedures or data needed to be checked or corrected. 

### 2.5. Statistical Analysis 

With a standard deviation (SD) of 8 in the total IGSQ score, a two-sided significance level of α = 0.05, a power of 90%, and a loss to follow-up rate of 15%, 91 infants were needed to detect a reduction of 3 scores in Day 14 compared with Day 0. Continuous variables were described using mean and SD, while categorical variables were described using frequency and proportion. Regarding the score of any question in IGSQ or simplified IGSQ that was recorded as “Unknown”, the median value of the remaining scores for the corresponding question was assigned. The proportion of the unknown was 1.0% for IGSQ and 3.6% for simplified IGSQ. Due to the repeated measurements, linear mixed models, where the duration of feeding intervention was used as a fixed effect and the individual participant as a random effect, were performed to examine the differences in various scores including the IGSQ score and the 5 domain scores among Day 0, Day 7, and Day 14. The same models were also performed to examine the gastrointestinal symptoms and sleeping hours that were related to the first 3 days of feeding intervention. Regarding the stool characteristics or the wake-up times at night that happened in the first 3 days, generalized linear mixed models were performed. The same models were also performed to examine the occurrence of digestive distress among Day 0, Day 7, and Day 14. Infants’ length, weight, head circumference, and BMI between Day 0 and Day 14, in original measurements and corresponding z-scores, were compared using Paired T tests. All statistical analysis and graphical illustration were completed using R software (Version 4.0.2). All statistical tests were two-sided with a significant level of α = 0.05.

## 3. Results

Among the 92 infants recruited, 12 were excluded for various reasons, and 80 (87%) completed the 14-day trial and remained in the final analysis (Figure 1). The demographic characteristics and IGSQ scores at baseline between the included and the excluded subjects were similar (Appendix A). The demographic characteristics of the 80 included infants are shown in Table 1. Mean (SD) maternal age was 32.0 (5.3) years, and 57.5% of infants’ mothers had college degrees. Male infants accounted for 48.8%. Mean (SD) birth weight and infants’ age were 3266 (470) g and 2.0 (1.5) months. All infants were fed by formula exclusively or partially. The median consumption of the study formula over the past 24 h at Day 7 and Day 14 was 450 mL and 480 mL, respectively, approximately accounting for 70% of what infants consumed.

The IGSQ score (SD) was reduced over a period of feeding intervention from 36.0 (5.7) at Day 0 to 28.7 (7.4) at Day 7 and to 26.5 (8.1) at Day 14 (*p* for trend <0.001). Compared with Day 0, the mean difference (95% confidence interval [95% CI]) for Day 7 was −7.3 (−9.0, −5.7) and −9.6 (−11.2, −7.9) for Day 14. The difference between Day 7 and Day 14, −2.2 (−3.9, −0.5), was also statistically significant (*p* = 0.010). The improvement persisted over the 14-day intervention for fussiness and crying, while it appeared more effective in the first week for vomiting and flatulence. Regarding stooling, relevant characteristics were not substantially changed during the two-week intervention (Table 2). Notably, the occurrence of digestive distress was significantly reduced during a period of feeding intervention (*p* for trend <0.001), 87.5% (*n* = 70) at Day 0, 35.0% (*n* = 28) at Day 7, and 28.8% (n = 23) at Day 14 (*p* for trend <0.001). Compared with Day 0, the odds ratio (OR) for Day 7 was 0.024 (95% CI: 0.005, 0.078) and 0.016 (95% CI: 0.003, 0.055) for Day 14; compared with Day 7, the OR for Day 14 was 0.652 (95% CI: 0.280, 1.467). Individual scores for vomiting, flatulence, and crying were significantly reduced in the first three days of feeding intervention (Table 3). Compared with Day 0, the reduction in scores for vomiting and flatulence at Day 1 was significant (*p* < 0.05), while it was not significant for crying until Day 2. Regarding fussiness and stool characteristics, no significant changes were found (Table 3). 

In the first three days, 22 (27.5%), 21(26.3%), and 17 (21.3%,) of infants were reported not to have defecated, respectively. Among infants who were reported to have defecated, the mean defecating times were 1.5, 1.5, and 1.6 from Day 1 to Day 3, respectively. Mushy or runny stools dominated. Stool consistency was not materially changed (Table 3), and sleeping hours and wake-up times during the night were not changed (Appendix A).

Noticeably, growth parameters in original measurements including length, weight, head circumference, and BMI were all increased significantly, but not for all z-scores. Z-scores for weight and BMI were significantly elevated (*p* < 0.001), while z-scores for length and head circumference were not significantly changed (*p* > 0.05) (Figure 2 and Appendix A).

Of the 80 parents, only one parent reported unsatisfactory with the feeding intervention, and no parents reported any adverse events during the study period.

## 4. Discussion

This single-armed, before and after trial assessed the effects of a pHF containing low lactose and probiotics on gastrointestinal functions and growth parameters in infants with MGDs. After the 2-week feeding intervention, the infants showed improved gastrointestinal functions except for stool characteristics and adequate growth. The improvement of gastrointestinal symptoms was manifested in the first week and even earlier for vomiting, flatulence, and crying. During the study period, no parents reported adverse events, while only 1 of 80 parents reported unsatisfactory with the feeding intervention.

Both total IGSQ scores and digestive distress prevalence were significantly decreased, particularly in the first week of the feeding intervention. Total IGSQ scores were reduced by 7.3% in the first week and by 2.2% in the second week, while digestive distress prevalence was correspondingly reduced by 60.0% and 17.9%. Similar findings were previously reported [1,17,21]. Pina et al. reported relieving MGDs in infants who were fed with low-lactose or lactose-free formula in 3–8 days [1]. Savino et al. reported a reduction of crying episodes in colic infants who were fed with a pHF containing oligosaccharides versus a standard formula for 14 days. The frequency of colic episodes in the pHF group was reduced by 58.8% in the first week and by 28.7% in the second week [22]. Consistent with total IGSQ scores, individual symptom scores except stooling were also significantly reduced after one week of feeding. Notably, the scores for three symptoms, including vomiting, flatulence, and crying, were reduced as early as the first three days of feeding intervention. Berseth et al. reported that the scores for spit-up, gas, fussiness, and crying were significantly reduced within one day when fussy infants were fed a pHF with low lactose [17], while Turco et al. reported crying was not improved in colic infants who were fed a pHF containing low lactose and Lactobacillus reuteri DSM 17,938 for 28 days [12]. Inconsistent results might be related to various factors, such as protein source, degree of protein hydrolysis, or lactose content [23,24].

Our study did not find the two-week feeding intervention materially changed stool characteristics consistent with a previous report that stool consistency was not improved in children who were fed a pHF for 28 days [17]. Our study did not find any improvement in sleeping hours in the first three days of feeding intervention, possibly due to our narrowed study duration. Our study, however, found growth parameters, including weight- and BMI-specific z-scores, were significantly improved. In particular, the mean weight-for-age z-score was noticeably elevated from the negative at baseline to the positive after the two-week intervention. This appears quite meaningful for infants with MGDs given that their growth velocity, without the feeding intervention, might not be changed that much as for infants without MGDs [4], and their z-scores should be relatively stable in such a short period of time. During the two-week intervention, no adverse events were reported in our study. A previous study, in which infants were fed a pHF containing Lactobacillus rhamnosus GG for one year and then followed up for 5 years, did not report any clinically relevant adverse events [25]. Recently, a systematic review with an expert consensus survey concluded that feeding non-exclusively breastfed infants a pHF is as safe as feeding an intact cow’s milk protein formula [24].

The underlying mechanisms for the effects of a pHF containing low lactose and probiotics on MGDs are not well known. There are several possible explanations. First, the formula likely downregulated allergic responses related to proteins or lactose. The IgG antibody was lower in infants fed a pHF versus those fed a standard formula [26], likely reducing allergic reactions. In addition, probiotics, especially bifidobacteria, can improve gastrointestinal functions. The colonic environment for infants fed a formula containing bifidobacteria is similar to that for breastfed infants, thus benefiting gastrointestinal health [13].

To the best of our knowledge, this was the first trial to evaluate the effects of a pHF containing low lactose and probiotics on gastrointestinal functions in infants with MGDs, though single-armed trials often took place in pediatric research [27,28,29,30]. Besides the full assessment of gastrointestinal functions by using IGSQ at Day 0, Day 7, and Day 14, we further collected information from Day 1 to Day 3 both on gastrointestinal functions using simplified IGSQ and on stool characteristics by stool diary. In addition, various measures were taken to ensure data quality, such as developing an operational manual, holding a training program, conducting a pilot, and monitoring progress online. Our study had limitations. First, randomized controls were not included due to ethical considerations, since a control arm would have restrict enrolled infants to switch formula, likely going against parents’ will [18]. Therefore, we cannot completely rule out a possibility that the improvement in infants’ gastrointestinal functions is related to a placebo effect, especially for the improvement in the first three days. In the future, a replication of these improvement effects using a cross-over design or follow-up after Day 14 would be merited. The improvement might be also related to infants’ maturity, whereas infant maturity in a very short period of time might not be evident. A previous study reported that a median IGSQ was reduced by five scores over a half year for infants ages 0–2 months [19], while our study found the median was reduced by as high as 10 scores only in two weeks for MGD infants who were fed a pHF. Second, feeding intervention lasted merely 2 weeks, likely limiting the generalization of the results to longer-term scenarios. Finally, gastrointestinal functions, such as in previous studies [18], were assessed mainly according to parents’ reporting rather than biochemical analysis.

## 5. Conclusions

In summary, the two-week feeding of a pHF containing low lactose and probiotics improves gastrointestinal functions in infants with MGDs. The improvement manifests in the first week of feeding, or likely even earlier in the first three days, particularly for symptoms of vomiting, flatulence, and crying. To understand the underlying biological mechanisms, such as the microbiota changes before and after feeding, further study is warranted.

## Figures and Tables

**Figure 1 nutrients-13-03371-f001:**
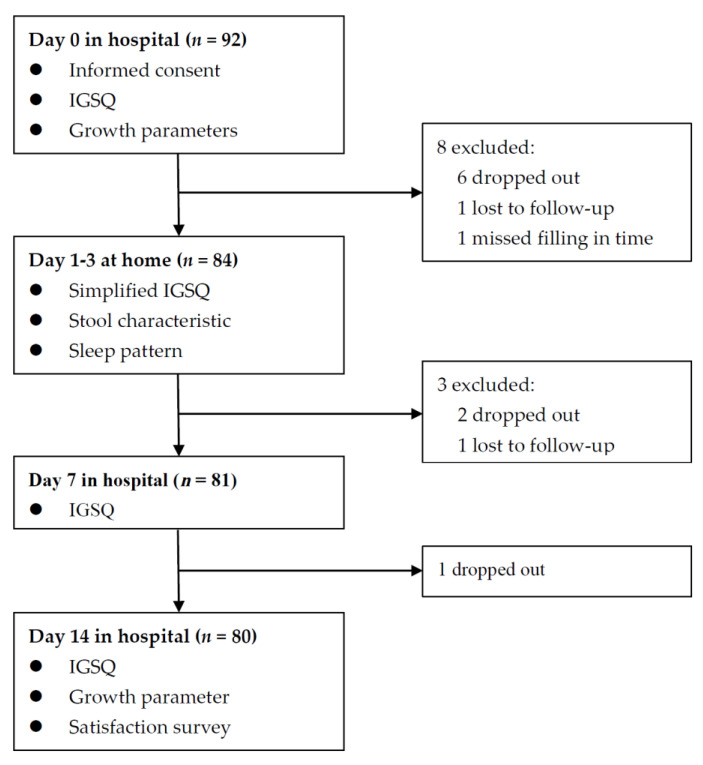
Flow chart of enrolled infants.

**Figure 2 nutrients-13-03371-f002:**
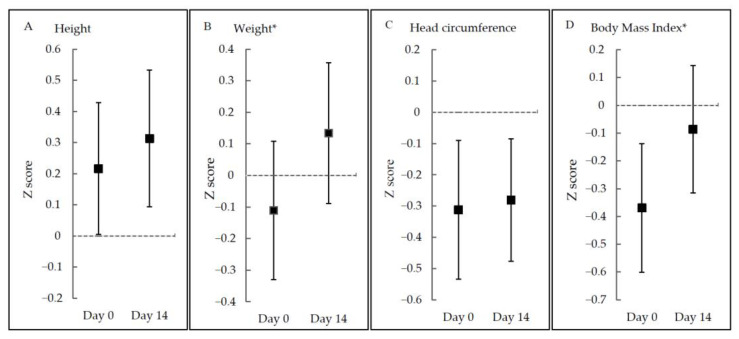
Z-scores at Day 0 and Day 14 according to growth parameters. * *p* < 0.05.

**Table 1 nutrients-13-03371-t001:** Baseline characteristics (*n* = 80).

Characteristics	Number (%)/Mean ± SD
Sex	
Male	39 (48.8)
Female	41 (51.3)
Age (months)	2.0 ± 1.5
Delivery mode	
Vaginal delivery	50 (62.5)
Cesarean delivery	30 (37.5)
Birth weight (g)	3266 ± 470
Feeding practice	
Mixed feeding	78 (97.5)
Exclusive formula feeding	2 (2.5)
Maternal age (year)	32.1 ± 5.3
Paternal age (year)	32.5 ± 5.8
Maternal education *	
College or above	46 (58.2)
Senior high school	25 (31.6)
Junior high school or below	8 (10.1)
Paternal education *	
College or above	48 (60.8)
Senior high school	23 (29.1)
Junior high school or below	8 (10.1)

Note. SD, standard deviation. * Education of one infant’s parents was unknown.

**Table 2 nutrients-13-03371-t002:** IGSQ scores at Day 0 (baseline), Day 7, and Day 14 of feeding intervention.

Domains (Score Range)	Day 0 (Baseline) Mean ± SD	Day 7		Day 14		
		Mean ± SD	MD (95% CI) vs. Day 0	Mean ± SD	MD (95% CI) vs. Day 0	MD (95% CI) vs. Day 7
Vomiting (4 to 20)	10.0 ± 2.9	7.9 ± 3.0	−2.2 (−2.8, −1.5) *	7.3 ± 2.9	−2.7 (−3.4, −2.0) *	−0.5 (−1.2, 0.1)
Flatulence (3 to 15)	7.7 ± 1.3	5.1 ± 1.9	−2.6 (−3.0, −2.1) *	4.8 ± 1.7	−2.9 (−3.3, −2.5) *	−0.4 (−0.8, 0.1)
Crying (2 to 10)	7.4 ± 2.7	6.4 ± 3.0	−1.0 (−1.7, −0.3) *	5.7 ± 2.7	−1.8 (−2.4, −1.1) *	−0.8 (−1.4, −0.1) *
Fussiness (2 to 10)	9.1 ± 2.1	7.6 ± 2.9	−1.6 (−2.2, −1.0) *	6.8 ± 2.8	−2.4 (−3.0, −1.7) *	−0.8 (−1.2, −0.2) *
Stooling (2 to 10)	4.1 ± 1.7	3.9 ± 2.0	−0.2 (−0.7, 0.3)	3.8 ± 1.9	−0.3 (−0.8, 0.2)	−0.1 (−0.6, 0.4)
Total scores (13 to 65)	36.0 ± 5.7	28.7 ± 7.4	−7.3 (−9.0, −5.7) *	26.5 ± 8.1	−9.6 (−11.2, −7.9) *	−2.2 (−3.9, −0.5) *

Note. IGSQ, Infant Gastrointestinal Symptom Questionnaire; MD, mean difference; SD, standard deviation; 95% CI, 95% confidence interval. * *p* < 0.05.

**Table 3 nutrients-13-03371-t003:** Simplified IGSQ scores and stool characteristics in the first 3 days of feeding intervention.

Domain	Day 0	Day 1		Day 2		Day 3	
	Mean ± SD/Frequency (%)	Mean ± SD/Frequency (%)	MD (95% CI)/ OR (95% CI)	Mean ± SD/Frequency (%)	MD (95% CI)/ OR (95% CI)	Mean ± SD/Frequency (%)	MD (95% CI)/ OR (95% CI)
Simplified IGSQ (score range)							
Vomiting (1 to 5)	3.6 ± 1.3	2.9 ± 1.3	−0.7 (−0.9, −0.5) *	2.7 ± 1.3	−0.9 (−1.1, −0.7) *	2.6 ± 1.3	−1.0 (−1.2, −0.7) *
Flatulence (1 to 5)	4.6 ± 0.8	2.8 ± 1.1	−1.8 (−2.0, −1.6) *	2.7 ± 1.1	−1.9 (−2.1, −1.7) *	2.7 ± 1.0	−1.9 (−2.1, −1.7) *
Crying (1 to 5)	2.4 ± 0.7	2.3 ± 1.1	−0.1 (−0.3, 0.1)	2.2 ± 1.0	−0.2 (−0.4, −0.03) *	2.0 ± 0.9	−0.4 (−0.6, −0.2) *
Fussiness (1 to 5)		3.3 ± 1.3		3.4 ± 1.3	−0.03 (−0.2, 0.3)	3.3 ± 1.4	−0.04 (−0.4, 0.2)
Stool Characteristic							
Pain or not							
Without pain		102 (82.9)	1.0 (reference)	90 (75.6)	1.4 (0.5, 4.0)	92 (74.2)	1.0 (0.4, 2.8)
With pain		21 (17.1)		29 (24.4)		32 (25.8)	
Consistency							
Watery stool		16 (13.0)	1.0 (reference)	9 (7.6)	0.99 (0.70, 1.40)	10 (8.1)	1.00 (0.71, 1.40)
Runny stool		52 (42.3)		57 (47.9)		62 (50.0)	
Mushy stool		50 (40.7)		50 (42.0)		44 (35.5)	
Firm stool		2 (1.6)		3 (2.5)		3 (2.4)	
Hard stool		3 (2.4)		0 (0.0)		5 (4.0)	

Note. IGSQ, Infant Gastrointestinal Symptom Questionnaire; MD, mean difference; OR, odds ratio; SD, standard deviation; 95% CI, 95% confidence interval. * *p* < 0.05.

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
