# Peer review of "The Effects of a Partially Hydrolyzed Formula with Low Lactose and Probiotics on Mild Gastrointestinal Disorders of Infants: A Single-Armed Clinical Trial"

_nutrients, 2021, doi:10.3390/nu13103371_

Round 1
Reviewer 1 Report
Open trial.
The fact that significant changes were noticed from Day 1 and 2 onward strongly suggest a pure placebo effect.
I miss a power calculation.
The conclusion of this paper is that this formula results in adequate growth and can be used as an alternative for standard infant formula. All the argumentation of therapeutic effect should be minimized, because of the extremely likelihood that this is pure placebo effect. But that is fine: infants grow well and parents are happy with an adequate formula.
Author Response
Dear reviewer,
nutrients-1385782, entitled “The effects of a partially hydrolyzed formula with low-lactose and probitics on mild gastrointestinal disorders of infants: A single-armed clinical trial”.
Thank you very much for your explicit guidance for our revision. We appreciate your comments. We have addressed all comments and revised the manuscript accordingly. All Changes in the manuscript were highlighted for easily reading. If there are any further questions or concerns, we will be happy to address them.
Sincerely yours,
Jian-meng Liu, MD, PhD.
Professor and Director
Ministry of Health Key Laboratory of Reproductive Health
Institute of Reproductive and Child Health
Peking University
Reviewer 1:
Comments and Suggestions for Authors
- Open trial. The fact that significant changes were noticed from Day 1 and 2 onward strongly suggest a pure placebo effect.
Response: Thank you for your helpful comments. We agree with you. The lack of a control group, due to ethical considerations, is a major limitation for a single-armed trial. We cannot completely rule out a possibility that the improvement in infants’ gastrointestinal functions is related to a placebo effect, especially for the improvement in the first three days. We added a sentence to address this limitation in discussion section (lines 275-278). We also rephrased sentences to make conclusion more conservative (line 31 in abstract, line 291 in main text).
- I miss a power calculation.
Response: The sample size calculation was added in Methods section: “With a standard deviation (SD) of 8 in the total IGSQ score, a two-sided significance level of α = 0.05, a power of 90%, and a lost to follow-up rate of 15%, 91 infants were needed to detect a reduction of 3 scores in Day 14 as compared with Day 0.” (lines 145-147).
- The conclusion of this paper is that this formula results in adequate growth and can be used as an alternative for standard infant formula. All the argumentation of therapeutic effect should be minimized, because of the extremely likelihood that this is pure placebo effect. But that is fine: infants grow well and parents are happy with an adequate formula.
Response: Thank you. The limitation about placebo effect was addressed in Discussion section (lines 275-278). We rephrased sentences to make conclusion more conservative (line 31 in abstract, line 291 in main text).
Reviewer 2 Report
the authors of the manuscript " the effects of a........clinical trial" has been executed and written very well. Also the authors did a great job of trying to check feeding of a pHF contaning low lactose and probiotic improves gastrointestinal functions in infants with MGDs. the improvement manifests in the first week of feeding. As the authors have rightly mentioned that underlying biological mechanisms are warranted in further study.
Only one suggestion would be if the authors can check the microbiota changes in pre and post study would be of great interest.
Author Response
Dear reviewer,
nutrients-1385782, entitled “The effects of a partially hydrolyzed formula with low-lactose and probiotics on mild gastrointestinal disorders of infants: A single-armed clinical trial”.
Thank you very much for your explicit guidance for our revision. We appreciate your comments. We have addressed all comments and revised the manuscript accordingly. All Changes in the manuscript were highlighted for easily reading. If there are any further questions or concerns, we will be happy to address them.
Sincerely yours,
Jian-meng Liu, MD, PhD.
Professor and Director
Ministry of Health Key Laboratory of Reproductive Health
Institute of Reproductive and Child Health
Peking University
Reviewer 2:
Comments and Suggestions for Authors
the authors of the manuscript " the effects of a........clinical trial" has been executed and written very well. Also the authors did a great job of trying to check feeding of a pHF containing low lactose and probiotic improves gastrointestinal functions in infants with MGDs. the improvement manifests in the first week of feeding. As the authors have rightly mentioned that underlying biological mechanisms are warranted in further study.
Response: We appreciate your comments.
- Only one suggestion would be if the authors can check the microbiota changes in pre and post study would be of great interest.
Response: Thank you very much for the good suggestion. We did not collect stool samples, but it is true that exploring microbiota changes in pre- and post-intervention is helpful to understand the underlying mechanisms. This is valued to be studied in future. We address this in line 293.
Reviewer 3 Report
In this open label study, authors evaluated the short term effects of a feeding formula consisting of partially hydrolyzed protein, low-lactose and probiotics on mild gastrointestinal disorders of infants measured by IGSQ. Results showed improved overall IGSQ scores at days 7 and 14, over day 0 and similar improvements in all sub scores except stooling. There is also some evidence of early appearance of the formula effects. There are significant study design draw backs that undermine the conclusions
- Open label design is the major issue because it can induce significant placebo effects to the care givers evaluating infants symptoms. While it can be argued weather double blind study design might be unethical or a barrier to recruit subjects, other study designs might had implemented to ameliorate the placebo effects e.g. cross over design, follow-up after day 14 and re-chalenge upon symptoms re-appearance.
- Which of the formula components is related to the effects or otherwise can you weight the effects of the three components of the formula? It is impossible with this study design. A 4 (or more) arms trial might give an answer, but it is complicated.
- Why did authors choose a short term trial and how can they compare their results with those of longer specific feeding duration?
- No sample size estimation
- No FU after the intervention
- Detection of significantly improved growth parameters including weight- and BMI-specific z-scores is arbitrarly related to formula feeding. Absence of controls does not permit this conclusion.
Author Response
Dear reviewer,
nutrients-1385782, entitled “The effects of a partially hydrolyzed formula with low-lactose and probiotics on mild gastrointestinal disorders of infants: A single-armed clinical trial”.
Thank you very much for your explicit guidance for our revision. We appreciate your comments. We have addressed all comments and revised the manuscript accordingly. All Changes in the manuscript were highlighted for easily reading. If there are any further questions or concerns, we will be happy to address them.
Sincerely yours,
Jian-meng Liu, MD, PhD.
Professor and Director
Ministry of Health Key Laboratory of Reproductive Health
Institute of Reproductive and Child Health
Peking University
Reviewer 3. Comments and Suggestions for Authors
In this open label study, authors evaluated the short term effects of a feeding formula consisting of partially hydrolyzed protein, low-lactose and probiotics on mild gastrointestinal disorders of infants measured by IGSQ. Results showed improved overall IGSQ scores at days 7 and 14, over day 0 and similar improvements in all sub scores except stooling. There is also some evidence of early appearance of the formula effects. There are significant study design draw backs that undermine the conclusions
Response: Thank you very much. We appreciate your comments. We have addressed all comments and revised the manuscript accordingly.
- Open label design is the major issue because it can induce significant placebo effects to the care givers evaluating infants symptoms. While it can be argued weather double blind study design might be unethical or a barrier to recruit subjects, other study designs might had implemented to ameliorate the placebo effects e.g. cross over design, follow-up after day 14 and re-chalenge upon symptoms re-appearance.
Response: Thank you very much for the good comments. We agree with you. A placebo effect is a main limitation in the study design. We have addressed this in Discussion section (Lines 273-278). We agree that other study designs such as cross over design, or follow-up after day 14, but due to various reasons such as budget restrictions, field implementation and parents’ compliances, a simple study design that is easily implemented with a high compliance has to be a choice. We have added sentences to clarify this in Discussion sections (lines 278-279).
Which of the formula components is related to the effects or otherwise can you weight the effects of the three components of the formula? It is impossible with this study design. A 4 (or more) arms trial might give an answer, but it is complicated.
Response: Thank you for the nice comments. The aim of our study is to evaluate the combined effects of the three components contained in the formula. It is correct that our study can not weight an individual component effect. Previous studies have studied the individual effect of partially-hydrolyzed protein formula (J Pediatr Gastroenterol Nutr 2014, 58, 549-552; Pediatrics 2002, 110, 1199-1203.), low lactose formula (J Pediatr 1999, 135, 587-592), and probiotics-containing formula on gastrointestinal symptoms in infants (Front Nutr 2018, 5, 39). To the best of our knowledge, no study has examined the individual effect of these three components in one population at the same time. We agree that a future 4-armed trial would be helpful.
- Why did authors choose a short term trial and how can they compare their results with those of longer specific feeding duration?
Response: We chose a short term trial, mainly because of budget restrictions, implementation feasibility and parents’ compliances. When designed the study, we also reviewed previous studies on infants’ gastrointestinal functions. The feeding duration in several studies was two weeks (â‘ Eur J Clin Nutr 2006;60:1304-10. â‘¡Pediatrics 1995;95:50-4. â‘¢Acta Paediatr Suppl 2003;91:86-90.).
It’s true that intervention periods played an important role, while a completely comparable study is not available. When making comparisons, key characteristics need to be considered, such as feeding intervention, outcomes, participants, or feeding duration. We just presented these characteristics of the previous studies to provide readers more details in the discussion section (lines 227, 230, 231, 236, 243, etc.).
- No sample size estimation
Response: The sample size estimation was added in Methods section: “With a standard deviation (SD) of 8 in the total IGSQ score, a two-sided significance level of α = 0.05, a power of 90%, and a lost to follow-up rate of 15%, 91 infants were needed to detect a reduction of 3 scores in Day 14 as compared with Day 0.” (lines 145-147).
- No FU after the intervention
Response: Due to budget restriction, we did not follow up children after intervention.
- Detection of significantly improved growth parameters including weight- and BMI-specific z-scores is arbitrarly related to formula feeding. Absence of controls does not permit this conclusion.
Response: We agree. Primary outcomes of our study are gastrointestinal functions, but growth parameters are routinely reported in formula evaluation study. We rephrased the summary of the main findings in the discussion section to make it more conservative: “After two-weeks’ feeding intervention, the infants showed improved gastrointestinal functions except stool characteristics and adequate growth.” (lines 217-219).
Round 2
Reviewer 1 Report
The authors addressed my comments but I still do find that this paper is not of sufficient quality for this journal
Reviewer 3 Report
Authors provided explanations for the major methodological caveats of their research. However, budget restrictions, implementation feasibility and parents’ compliances are obstacles that significantly lower the value of the study and cannot be an excuse for a better study design.